# Disordered Eating Behaviours and Eating Disorders in Women in Australia with and Without Polycystic Ovary Syndrome: A Cross-Sectional Study

**DOI:** 10.3390/jcm8101682

**Published:** 2019-10-14

**Authors:** Stephanie Pirotta, Mary Barillaro, Leah Brennan, Angela Grassi, Yvonne M. Jeanes, Anju E. Joham, Jayashri Kulkarni, Lynn Monahan Couch, Siew S. Lim, Lisa J. Moran

**Affiliations:** 1Monash Centre for Health Research and Implementation, Monash University, Melbourne, VIC 3168, Australia; anju.joham@monash.edu (A.E.J.); siew.lim1@monash.edu (S.S.L.); 2Faculty of Health Sciences, School of Behavioural and Health Sciences, Australian Catholic University, Melbourne, VIC, Australia; Centre for Eating, Weight and Body Image, Melbourne, VIC 3065, Australia; mary.barillaro@myacu.edu.au (M.B.); leah.brennan@cewbi.com.au (L.B.); 3Nutrition Department, West Chester University of Pennsylvania, West Chester, PA 19383, USA; agrassi@pcosnutrition.com (A.G.); lmonahan@wcupa.edu (L.M.C.); 4Health Sciences Research Centre, Department of Life Sciences, University of Roehampton, London SW15 5PJ, UK; Y.Jeanes@roehampton.ac.uk; 5Department of Diabetes, Monash Health, Melbourne, VIC 3168, Australia; 6Monash Alfred Psychiatry Research Centre (MAPrc), Melbourne, VIC 3004, Australia; jayashri.kulkarni@monash.edu

**Keywords:** polycystic ovary syndrome, disordered eating, eating disorder, binge-eating

## Abstract

Psychological co-morbidities common in polycystic ovary syndrome (PCOS) may contribute to disordered eating and subsequent weight gain. This cross-sectional study aimed to determine the prevalence of disordered eating and a range of eating disorders and demographic risk factors associated with these behaviours within an Australian group of women with and without PCOS. Data from 899 women with (*n* = 501) and without (*n* = 398) PCOS were analysed as possibly indicative of disordered eating or eating disorders using the Eating Disorder Examination Questionnaire (EDE-Q) and The Diagnostic and Statistical Manual of Mental Disorders Fifth Edition (DSM-5) criteria. Disordered eating (*p* = 0.012) but not eating disorders (*p* = 0.076) were more prevalent in women with PCOS compared to controls. Increased body mass index (BMI) [Odds Ratio (OR): 1.03; 95%; Confidence Interval (CI): 1.01, 1.05, *p* = 0.012] and older age [OR: 1.05; 95%CI: 1.02, 1.08, *p* = 0.002] but not PCOS diagnosis [OR: 1.43; 95%CI: 0.96, 2.13 *p* = 0.078] increased the odds of disordered eating. Increased BMI [OR: 1.04; 95%CI: 1.02, 1.06, *p* < 0.001] and younger age [OR: -0.95; 95%CI: 0.93–0.95, *p* < 0.001] but not PCOS diagnosis [OR: 1.38; 95%CI: 0.97, 1.95, *p* = 0.076] increased the odds of an eating disorder. Clinicians are recommended to screen all women with PCOS for possible disordered eating behaviours, with particular attention to women with elevated BMI.

## 1. Introduction

Polycystic ovary syndrome is one of the most common endocrinopathies, affecting 8–13% of women [1] and is associated with a variety of reproductive [2], metabolic [3] and psychological symptoms. Of the psychological outcomes, women are particularly vulnerable to heightened perceived stress, body dissatisfaction, low self-esteem, disordered eating, anxiety and depression [4,5,6]. Factors such as infertility, hyperandrogenism, elevated body mass index (BMI), metabolic disorders and poor self-esteem have been suggested as possible contributors to adverse psychological outcomes among this population [6,7].

Eating disorders are psychological disorders characterised by abnormal or disturbed eating behaviours with or without compensatory behaviours as diagnosed by DSM-5 [8]. Behaviours include food restriction, binge-eating, purging, laxative use, diet pills and excessive exercise. Disordered eating is a condition characterised by the same characteristics of a lesser frequency or a lower level of severity as that of an eating disorder [8]. Disturbed eating behaviours have been found to be related to low self-worth or self-esteem primarily based on body weight or shape, disturbed experience of one’s own body, and anxiety towards particular foods and food groups [9]. The EDE-Q is the recommended clinical gold standard screening tool to assess the presence of key cognitive and behavioural features present in both disordered eating and eating disorders, highlighting the need for further psychological assessment. 

Women with PCOS have been reported to have a higher prevalence of both disordered eating behaviours [10,11,12] and eating disorders [6,13] compared to women without PCOS in the majority of but not all [14,15,16,17] prior literature. This includes a meta-analysis reporting an odds ratio of 3.05 for women with PCOS having disordered eating behaviours [12]. Of abnormalities in disordered eating behaviours, binge eating is of particular concern, being exhibited in 58% of women with PCOS compared to 32% of controls [18]. The literature examining eating disorders has predominantly focused on particular eating disorders such as binge eating disorder [19] or bulimia nervosa [13,20], with only a few studies assessing a variety of eating disorders [6,21,22,23]. There is, therefore, a lack of research examining the prevalence of disordered eating and the range of eating disorders in women with PCOS, particularly in an Australian cohort. 

There is also currently a lack of understanding as to why women with PCOS may experience greater rates of disordered eating behaviours and eating disorders compared to women without PCOS. Within the general population, risk factors are known to be diverse and broadly associated with psychosocial, demographic, genetic and environmental influencers [24,25,26]. A focus on demographic factors finds younger age to be associated with binge eating disorder and bulimia nervosa whilst obesity to be more common among people with eating disorders [27]. Immigrants have also been reported to have a reduced risk of eating disorders compared to non-immigrant populations [28]. Within a PCOS context, current literature focuses on psychological comorbidities such as low self-esteem and severe psychological distress as plausible risk factors for the development of disordered eating and eating disorders [6,12,15,29]. There are, however, currently no studies identifying demographic factors that may link PCOS to disordered eating or eating disorders, including country of birth, Indigenous (Aboriginal or Torres Strait Islander) status, BMI or age. Understanding these factors may make it easier for clinicians to identify the clinical profile of women with PCOS most at risk of eating disorders or disordered eating in practice.

The aim of the present study was to determine the prevalence of disordered eating and eating disorders among women with and without PCOS in Australia. A secondary aim was to determine the demographic factors associated with disordered eating and eating disorders in this population. It is therefore hypothesized that Australian women with PCOS will report significantly greater disordered eating and a range of eating disorders compared to women in the general population. It is also hypothesized that elevated BMI and younger age will be associated with disordered eating and eating disorders in all women, irrespective of PCOS diagnosis. 

## 2. Methods

### 2.1. Participants and Eligibility Criteria

To meet inclusion criteria, women in the PCOS group must have been 18–45 years old, be diagnosed with PCOS and fluent in the English language. Women were excluded if they were pregnant, breastfeeding or taking weight-loss medication up to 6 months prior to completing the questionnaire. Informed consent was received prior to initiating the survey. The Human Research Ethics Committees of Monash University approved the study methods, Human Resource Ethics Committee (HREC) Reference Number: AU/13/2BF2212. Of the controls, people were deemed eligible if they self-reported as Australian and were aged 18 years and older. Within this group, participants were sampled to be included as general population controls if they were female, aged 18–45 years and had completed all of the EDE-Q. Information regarding PCOS status was not collected from the controls. Informed consent was received prior to initiating the survey. Ethical approval for this study was obtained from the Australian Catholic University Ethics Committee, granted under HREC number 2017-79E.

### 2.2. Data Collection

Data for this online observational cross-sectional study was collected between August 2017 and March 2018. Women with PCOS (n = 501) were recruited through PCOS-related Australian social media pages as well as Australian e-newsletter distribution servers. Controls (*n* = 398) were sampled from a cross-sectional online study that aimed to examine a range of eating-related psychosocial variables within an Australian sample. Participants were recruited through social media and University intranet pages and groups between July and August 2017.

### 2.3. Outcome Measures

Women were questioned about their demographics, including date of birth, country of birth (Australia or countries other than Australia), Indigenous status (Aboriginal or Torres Strait Islander), self-reported PCOS diagnosis (determined using the question, ‘Have you ever been diagnosed with PCOS?’) and self-reported height and weight. BMI (based on self-reported height and weight) was calculated and grouped according to each category defined as “underweight”: <18.5kg/m^2^, ‘healthy weight”: ≥18.5–24.9kg/m^2^, “overweight”: ≥25.0 -29.9kg/m^2^ and “obese”: ≥30kg/m^2^ [30].

#### 2.3.1. Eating Disorder Examination Questionnaire

All participants completed the EDE-Q version 6 which is a self-reported 28-item instrument that measures the range and severity of eating disorder behaviours and cognitions including the frequency of self-reported key eating and compensatory behaviours (objective binge-eating episodes, self-induced vomiting, laxative use and excessive exercise) over the previous 28 days [31]. Cognitive responses were rated on a 7-point Likert scale whilst behavioural items were open-answers, with higher scores indicating greater severity. Answers are used to calculate the EDE-Q sub scores according to cognitive features of eating disorders (dietary restraint, eating concern, weight concern and shape concern) over the previous 28 days. The average of the sub scores was then used to calculate the final overall EDE-Q global score (Fairburn et al. [32]). Global scores and sub scores ≥4 were considered clinically indicative disordered eating behaviours requiring further psychological assessment through clinical interview [31]. Behavioural items were assessed according to frequency of occurrence in the last 28 days and were not used to compute subscale scores. Scores were not used to directly indicate an eating disorder or disordered eating but rather provide the clinician with descriptive information to better understand the manifestation of these behaviours [23,33]. The EDE-Q has been shown to have a sensitivity rating of 83% and specificity of 96% [34].

#### 2.3.2. Disordered Eating Criteria

Disordered eating was classified as a global EDE-Q score ≥ 1.52 (the community average) [35] and the absence of any eating disorder diagnosis according to DSM-5 criteria. Following standard practice when using the EDE-Q, frequencies reported over the last 28 days were considered indicative of the previous 3 months as the EDE-Q only assesses the previous 28 days and the DSM-5 requires the presence of eating disorder behaviours to have taken place within the last 3 months [33]. 

#### 2.3.3. Eating Disorder Screening Criteria

All potential eating disorder diagnosis, apart from anorexia nervosa, were determined using the American Psychiatric Association criteria [8], with criteria corresponding to particular questions within the EDE-Q in addition to clinical features where required. Anorexia nervosa was diagnosed with a low body weight (BMI <17.5kg/m^2^) to aid analysis as DSM-5 does not offer a BMI cut-off point (Table A1). The frequency of key eating and compensatory behaviours were used to identify eating disorder diagnosis (Table A1). Dietary restraint as an eating disorder criteria was measured using item 1 of the EDE-Q, ‘How many times in the last 28 days have you deliberately tried to limit the amount of food you eat to influence your shape or weight (whether or not you have succeeded?)’. Frequent occurrence of dietary restraint was defined as long periods of time (8 hours or more) without eating anything in order to influence shape or weight >3 per week in the past 28 days [23]. Frequent occurrence of excessive exercise was defined as exercising in a driven or compulsive way as a means of controlling weight, shape or amount of fat, or to burn calories or ≥20 times over the past 28 days (item 18) [23]. All other compensatory behaviours (objective binge-eating (item 15), self-induced vomiting (item 16) and laxative use (item 17)) were considered regular with an occurrence of ≥4 over the past 28 days [33]. 

Anorexia nervosa was defined as BMI ≤ 17.5kg/m^2^ and trying to deliberately limit food intake to influence weight or shape (item 1 ≥1) as well as either weight (item 22 ≥4) or shape (item 23 ≥4) influencing personal self-judgement. Binge-eating disorder was classified as the presence of objective binge-eating (item 15 ≥4) with no other compensatory behaviours, whilst bulimia nervosa was classified as objective binge eating (item 15 ≥4) with at least one compensatory behaviour occurring at least once per week over the last 28 days (item 16, 17 or 18 ≥4). Other specified feeding or eating disorders were classified as having atypical anorexia nervosa, low frequency bulimia nervosa, low frequency binge-eating disorder or purging disorder. Atypical anorexia nervosa met the same anorexia nervosa guidelines but had a BMI >17.5kg/m^2^, low frequency binge-eating and low frequency bulimia nervosa differed due to a reduced frequency of compensatory behaviours (items 16 or 17 or 18 score 1–3). Purging disorder was classified as self-induced vomiting (item 16 ≥4), laxative use (item 17 ≥4) and dietary restraint (item ≥3) with no other eating disorders. Unspecified feeding or eating disorder was classified as a global score ≥4 with no other eating disorders diagnosed.

### 2.4. Statistical Analysis

All analyses were carried out using SPSS version 25 2017 (IBM, Armonk, NY, USA). Descriptive findings for continuous data were reported using means and standard deviations and median and interquartile ranges for non-normally distributed data sub scores. Descriptive findings for categorical data (proportion of women meeting disordered eating or eating disorder criteria or the proportion reporting clinically indicative sub scores) were reported using frequencies. Mean differences between PCOS and controls were determined using an independent sample t-test or Mann-Whitney test for parametric and non-parametric continuous outcomes respectively. Differences between categorical variables was analysed using chi-square test or Fisher’s exact test when five or less cases were present. Binary logistic regression analysis was performed to examine the relationship between disordered eating or eating disorder (yes/no) and the independent variables: PCOS status (binary), age (continuous), BMI (continuous), country of birth (categorical) and indigenous status (binary). The selection of variables in the multivariable models were based on identifying variables suspected or known to impact the outcomes of interest and/or exhibited *p* < 0.1 upon univariable analysis. Statistical significance was defined as *p* < 0.05.

## 3. Results

### 3.1. Demographics

A total of 899 women (501 PCOS and 398 controls) with an average age of 27.1 ± 6.9 years, weight of 80.8 ± 25.5kg and BMI of 29.5 ± 9.2kg/m^2^ were included in this study. The sample of women with PCOS was significantly older, had higher BMI, had a greater portion of Indigenous or Torres Strait Islander people, was more likely to be from European descent and less likely from Asia when compared to controls (Table 1). The majority of women with PCOS were regarded as having obesity whilst controls were mostly within the healthy weight range. 

### 3.2. Clinically Indicative EDE-Q Scores

The severity of overall and individual eating disorder psychopathologies measured using the EDE-Q global score and sub scores respectively among women with and without PCOS are outlined in Table 2. Dietary restraint was the only psychopathology significantly more common among women with PCOS. A greater portion of controls had a clinically indicative global score, weight and shape concern. When comparing means, however, women with PCOS had significantly elevated mean global scores when compared to controls (2.30 vs. 2.08, *p* = 0.016) (Table A2). 

### 3.3. Prevalence of Disordered Eating and Eating Disorder Diagnosis

The prevalence of disordered eating and eating disorders is outlined in Table 2. Disordered eating behaviour was significantly more common amongst women with PCOS compared to controls (21 vs. 15%, *p* = 0.012). Eating disorder diagnosis was not significantly different between women with and without PCOS (62 vs. 56%, *p* = 0.076). Anorexia nervosa (0% vs. 1.3%, *p* = 0.012) and atypical anorexia nervosa (0% vs. 5.8%, *p* = <0.001) were not diagnosed among women with PCOS and hence significant differences in prevalence were observed between the groups.

### 3.4. Demographic Predictors of Disordered Eating and Eating Disorder Diagnosis

As outlined in Table 3, only PCOS diagnosis, older age and elevated BMI were associated with increased odds of disordered eating upon univariable analysis. When adjusting for these in multivariable analysis, women with PCOS were not at increased odds of disordered eating compared to women without PCOS. Rather older age and higher BMI were independently associated risk factors for disordered eating behaviour. The demographic risk factors associated with eating disorder diagnosis are outlined in Table 4. When using univariable analysis, PCOS diagnosis, older age and elevated BMI increased the odds of an eating disorder. Yet, when adjusted for age and BMI, the odds ratio of an eating disorder was no longer significantly different between women with and without PCOS. Rather younger age and elevated BMI remained significant risk factors in the multivariable model.

## 4. Discussion

This paper is the first to assess both disordered eating and eating disorder prevalence and their demographic predictors among women with and without PCOS in Australia. We also report here the largest cross-sectional study available assessing the prevalence of these conditions using the validated EDE-Q screening questionnaire and DSM-5 criteria. Disordered eating behaviour, but not eating disorders, were more prevalent in women with PCOS. Of the core factors of eating disorders and disordered eating, dietary restraint was the only behaviour experienced more often in women with PCOS. Interestingly PCOS diagnosis was not a predictor for disordered eating behaviour or eating disorder diagnosis. Rather older age and higher BMI, irrespective of PCOS status, increased the odds of women taking part in disordered eating behaviour, whilst younger age and elevated BMI increased the risk of women being diagnosed with an eating disorder. 

In the majority of published literature, women with PCOS are reported to have a higher prevalence of disordered eating compared to controls [10,11,12,23]. A prior meta-analysis reported an odds ratio (OR) of 3.05 for disordered eating behaviour in women with PCOS compared to those without [12]. The current study showed a more modest difference in disordered eating between groups (20% vs. 15%), which may be attributed to the use of differing disordered eating and eating disorder criteria and screening tools between studies. Future research should use psychiatric interviews (e.g. EDE-Q) to identify eating disorder and disordered eating diagnosis [12]. The etiology of disordered eating among women with PCOS is still unclear but is thought to be related to several factors including higher BMI [36], weight and eating concerns [23], greater body dissatisfaction [37], depression and anxiety [38], poorer life quality [39] and higher dietary restraint [38]. 

We report here no differences in eating disorder prevalence between women with and without PCOS. Studies assessing eating disorders provide mixed results with some [6,12,13,20] but not all [14,15,16] reporting a higher prevalence for women with compared to those without PCOS. A meta-analysis reported the odds of being diagnosed within an eating disorder among women with PCOS were 3.87 higher than women in the general population [12]; however, again, this review included studies utilising a variety of eating disorder criteria and screening tools, reporting an overall moderate heterogeneity. Our use of the validated EDE-Q and internationally-recognised clinical eating disorder criteria may explain these discrepant results. We also extend previous research by examining a range of eating disorders in PCOS. We confirm prior reports that binge-eating disorder was the most prevalent eating disorder in PCOS [40], which is proposed to be likely as a result of an interplay between hormonal, psychological and metabolic influences [40]. Binge-eating disorder is also the most prevalent in the general population and as such results may also be a reflection of this [27]. As previously reported [23], anorexia nervosa and atypical anorexia nervosa were not diagnosed in women with PCOS. Lower levels of testosterone have been observed with anorexia nervosa [41]. As androgens have been proposed to act as appetite-stimulants, induce binge-eating behaviours and impair impulse control [42], the elevated levels of testosterone commonly observed in PCOS may explain the more common observance of binge-eating behaviours in comparison to anorexia nervosa [11,40]. 

The association here between higher BMI and both disordered eating and eating disorders is consistent with prior literature in the general population [43]. Higher BMI can lower body esteem and body satisfaction as well as alter eating behaviour [15], with these factors also known to contribute to disordered eating, particularly binge-eating [29,40]. Hormonal dysregulation impacting brain function may also be associated with higher BMI [44], with increasing fat mass influencing leptin and insulin resistance, suppressing satiety signals and permitting the over-consumption of food [45,46,47]. The elevated prevalence of overweight and obesity in PCOS [48] may therefore be a contributing factor to the higher occurrence of disordered eating in PCOS, rather than PCOS itself. This may explain the lack of an independent relationship between PCOS status and disordered eating.

As previously reported [49], younger age was associated with a higher risk of eating disorders, irrespective of PCOS diagnosis. Societal norms, close peer attitudes and frequent social media use are thought to contribute to increased body dissatisfaction and appearance-ideal internalisation in women. Furthermore, constant comparison of one’s body to the ideal societal slim physique may bring on feelings of shame, anxiety and sadness [50]. Conversely, older age was here associated with a higher risk of disordered eating in all women. Longitudinal data in the general population report that disordered eating attitudes and behaviours occur well into middle and later age [51], which may be related to factors such as perfectionism [52] and body dissatisfaction bought on by social norms or appearance-related critism from partners [51]. Life events such as pregnancy or menopause are thought to be major contributors to poor-self-image due to related weight gain. Dieting practices may then evince to conform to the idealistic thin female body perceived to be sexually desirable and of good health [53], posing as risk factors for disordered eating [54]. 

We also report here on psychopathologies common within the spectrum of eating disorders, including dietary restraint as a prospective risk factor for the development of disordered eating and eating disorders involving binge-eating behaviours [40]. Strategies such as calorie counting, limiting energy-dense foods and portion control are frequently incorporated into weight loss programs and are thought to be a successful predictor for weight loss in the general population [55]. However, an exaggerated preoccupation with dietary restraint may lead to long-term weight regain, increased stress [56], obsessive thoughts relating to forbidden foods [57], increased appetite sensations [55] and overeating related to disinhibition [58]. 

The PCOS International Evidence-Based guidelines [59] recognise lifestyle therapy as first-line treatment for the management of PCOS. Given the higher prevalence of disordered eating [12], associated psychopathologies and increased levels of stress in women with PCOS [6,43], lifestyle management plans should shift the focus from dietary restraint to other alternatives such as sustainable behaviour change techniques, self-efficacy and health-centered nutrition recommendations alongside psychological therapies. This will help avoid feelings of deprivation or restriction [60,61] and improve body image and self-esteem in order to promote long-term weight management, reduce dietary disinhibition, promote self-control and prevent the exasperation of disordered eating behaviours [59]. 

This study has a number of strengths. We report here the largest cross-sectional study to date assessing PCOS, disordered eating and eating disorders. We also assessed eating disorders by using the validated EDE-Q and internationally-recommended DSM-5 criteria. As controls were not screened for PCOS diagnosis, this group will also likely include some women with PCOS. We also note several limitations. We were unable to formally diagnose patients using a clinical psychological interview [62] and therefore the results presented in this paper need to be considered in this context. An interview is required to confirm the presence of disordered eating or eating disorders. Secondly, the cross-sectional nature of this study makes it impossible to assess the causality of eating disorders with further prospective research required. Psychological history was also not obtained and unable to be included in the analysis. Data regarding weight loss medication amongst controls was not obtained. This could potentially contribute to group differences in disordered eating behaviour given prior associations with either prescribed or non-prescribed weight loss medication and disordered eating and eating disorders [63,64]. The voluntary participation of this study may have also lead to selection bias and the ‘healthy worker effect’ [65]. PCOS diagnosis and weight were also self-reported and lacked clinical confirmation. Controls were also largely recruited through a tertiary setting which may not be representative of the general population as they have higher eating disorder prevalence (8–17%) [33] compared to the general population (4–16%) [27]. 

## 5. Conclusions

This is the largest cross-sectional study to date assessing PCOS, disordered eating and a range of eating disorders, whilst utilizing the validated EDE-Q and internationally-recognized DSM-5 criteria. Australian women with PCOS were found to have a higher prevalence of disordered eating but not eating disorders when compared to women without PCOS. Increased BMI and older age were found to increase the risk of disordered eating whilst increased BMI and younger age were associated with eating disorder diagnosis. Binge-eating disorder was the most commonly reported eating disorder in women with PCOS. We note limitations including that clinical psychological interviews were not used, psychological history was not obtained and weight and PCOS diagnosis were self-reported and lacked clinical confirmation. Clinicians are recommended to screen all women with PCOS for possible disordered eating behaviours, with particular attention to women of older age and elevated BMI. 

## Figures and Tables

**Table 1 jcm-08-01682-t001:** Participant baseline characteristics.

Variable	PCOS N = 501	Control N = 398	*p*-value
Age (years)	30.5 ± 5.9	22.8 ± 5.5	<0.001
Aboriginal or Torres Strait Islander	16 (3.2)	1 (0.3)	<0.001
*Country of birth*			0.003
Australia	432 (86.2)	330 (82.9)	0.170
Asia	15 (3.0)	25 (6.3)	0.018
Europe	26 (5.2)	14 (3.5)	0.003
Other	28 (5.6)	22 (5.5)	0.968
Weight (kg)	91.8 ± 25.5	66.9 ± 17.6	<0.001
BMI (kg/m^2^)	33.6 ± 9.3	24.3 ± 6.0	<0.001
Underweight (<18.5 kg/m^2^)	5 (1.0)	29 (7.3)	0.175
Healthy weight (18.5 - <25kg/m^2^)	90 (18.0)	233 (58.5)	0.025
Overweight (25 - <30 kg/m^2^)	88 (17.6)	80 (20.1)	0.001
Obese (≥30kg/m^2^)	312 (62.3)	49 (12.3)	0.049

Abbreviations: BMI, body mass index; PCOS, polycystic ovary syndrome. Differences between groups assessed using independent samples t-test or chi-square test for continuous or categorical variables respectively. Data are expressed as mean ± SD, n (%).

**Table 2 jcm-08-01682-t002:** EDE-Q survey results for women with and without PCOS.

Variable	PCOS N = 501	Controls N = 398	*p*-value
Prevalence of clinically indicative EDE-Q scores (≥4)	N (%)	N (%)	
Global score	36 (7.2)	53 (13)	0.002
Dietary restraint	88 (18)	42 (11)	0.003*
Weight concern	46 (9.2)	103 (26)	<0.001
Shape concern	85 (17)	127 (32)	<0.001
Eating concern	47 (9.4)	28 (7.0)	0.207
Prevalence of eating disorder diagnosis			
All eating disorder diagnosis	310 (62)	223 (56)	0.076
Anorexia nervosa	0 (0.0)	5 (1.3)	0.012
Bulimia nervosa	26 (5.2)	18 (4.5)	0.645
Binge-eating disorder	143 (29)	92 (23)	0.066
Other specified feeding or eating disorder	135 (27)	96 (24)	0.336
Atypical anorexia nervosa	0 (0.0)	23 (5.8)	<0.001
Low frequency bulimia nervosa	47 (9.4)	32 (8.0)	0.481
Low frequency binge-eating disorder	82 (16)	47 (12)	0.053
Purging disorder	6 (1.2)	3 (0.8)	0.507
Unspecified feeding or eating disorder	6 (1.2)	12 (3.0)	0.053
Disordered eating	107 (21)	59 (15)	0.012

Abbreviations: BMI, body mass index; EDE-Q, eating disorder examination questionnaire; PCOS, polycystic ovary syndrome. Differences between groups assessed using chi-square test or Wilcoxon-Mann-Whitney test where indicated (*).

**Table 3 jcm-08-01682-t003:** Factors associated with disordered eating in women with and without PCOS.

	Univariable					Multivariable				
Variable	β	SE	OR	*p*	95% CI [Lower, Upper]	β	SE	OR	*p*	95% CI [Lower, Upper]
PCOS status	−0.45	0.18	0.64	0.013	0.45, 0.91	0.16	0.23	1.17	0.477	0.75, 1.83
Age	0.06	0.01	1.06	<0.001	1.03, 1.08	0.05	0.02	1.05	0.002	1.02, 1.08
BMI	0.04	0.01	1.04	<0.001	1.02, 1.05	0.03	0.01	1.03	0.012	1.01, 1.05
COB	−0.33	0.26	0.72	0.207	0.43, 1.20	-	-	-	-	-
Indigenous and Torres Strait Islander	0.62	0.54	1.87	0.247	0.65, 5.37	-	-	-	-	-

Abbreviations: BMI, body mass index; COB, country of birth; OR, odds ratio; PCOS, polycystic ovary syndrome; SE, standard error; CI, confidence interval. Data are presented as odds ratio (OR) (95%CI) and were analysed using binary logistic regression.

**Table 4 jcm-08-01682-t004:** Factors associated with eating disorders in women with and without PCOS.

	Univariable					Multivariable				
Variable	β	SE	OR	*p*	95% CI [Lower, Upper]	β	SE	OR	*p*	95% CI [Lower, Upper]
PCOS status	0.24	0.14	1.27	0.077	0.98, 1.67	0.32	0.18	1.38	0.076	0.97, 1.95
Age	−0.02	0.01	0.98	0.064	0.96, 1.00	−0.05	0.01	0.95	<0.001	0.93, 0.97
BMI	0.03	0.01	1.03	<0.001	1.01, 1.04	0.04	0.01	1.04	<0.001	1.02, 1.06
COB	−0.15	0.19	0.86	0.425	0.60, 1.24	-	-	-	-	-
Indigenous and Torres Strait Islander	−0.08	0.17	0.93	0.647	0.66, 1.29	-	-	-	-	-

Abbreviations: BMI, body mass index; COB, country of birth; OR, odds ratio; PCOS, polycystic ovary syndrome; SE, standard error; CI, confidence interval. Data are presented as odds ratio (OR)(95%CI) and were analysed using binary logistic regression.

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
