# Peer review of "Disordered Eating Behaviours and Eating Disorders in Women in Australia with and Without Polycystic Ovary Syndrome: A Cross-Sectional Study"

_jcm, 2019, doi:10.3390/jcm8101682_

Round 1
Reviewer 1 Report
Summary
The aims of this study were to assess the prevalence of disordered eating and eating disorders among women with PCOS and to determine demographic factors associated with them in an Australian cohort of women with PCOS and controls from the general population. It was found that BMI and age were associated with disordered eating and diagnoses of eating disorders. PCOS diagnosis was not statistically associated, however the ORs were elevated for disordered eating and diagnoses of eating disorders (1.43 and 1.38, respectively), and the possibility of a too small sample size to detect the association should be considered. Strengths include the collection of data on a variety of disordered eating behaviors, and a well written manuscript. Limitations include various sources of bias which should be further acknowledge.
General comments
- Women in the PCOS group who were taking weight loss medication were excluded, but this was not mentioned as an exclusion criteria for the controls. Please discuss how this could bias your sample and whether this could be be responsible for the larger number of women in the control group who had clinically indicative EDE-Q scores in Table 2?
-The EDE-Q asks for responses based on the ‘previous 28 days’. If younger women are more likely to have eating disorders, than this may be a reason for why you found a higher rate in younger women. If you had asked about lifetime prevalence of eating disorders, do you anticipate that you would have still found a difference between younger and older women in the study population?
-Were the controls screened for PCOS and excluded if they reported having PCOS? This should be made clear. To be truly reflective of the general population, they should be selected without their PCOS status in mind.
- One acknowledged limitation is that PCOS is self-reported (it should be noted that weight is self-reported as well). Another limitation is that other sources of bias may be present since women with PCOS were volunteered to participate.
Specific points
Abstract – change the end of the last sentence to ‘…, with particular attention to women with elevated BMI.’
Lines 58-59 – In reference 18, I could not find where they state that binge eating was found in 50% of women with PCOS. Please check the reference again.
Table 2 – footnote: “*Data expressed as median and IQR” does not seem relevant for this table
Author Response
Dear reviewer,
Please see the attachment.
Regards
Stephanie Pirotta

Reviewer 2 Report
During a cross-sectional study, the authors planned to determine the prevalence of disordered eating and a range of eating disorders and demographic risk factors associated with these behaviors within an Australian cohort of women with and 24 without PCOS. Although the paper is interesting, the quality is not enough to published in JCM. The main concern of the present study is its methodology.
Here you can find my comments that suggest to be considered in revised version:
Title:
Word of “cross-sectional analysis” in the title can removed or change to cross-sectional study.
Abstract:
Aim of the study is unclear; the authors should rewrite it for better understand. Word of “cohort” should remove from abstract, since this study has a cross-section design. Results section is confusing. This section should be rewrite. The authors should focus on multiple logistic regression results as following sentence: The odds ratio of eating disorder adjusted for age and BMI in PCOS patients was not significantly different compared to those without PCOS.
Introduction
Introduction is very long. Please summarize it. The authors should describe related factors with disordered eating behaviors and eating disorders.
Materials and Methods
Paper has many subheadings; please remove unnecessary subheading. Subheading of “experimental section” is wrong since this study is a cross-sectional study. Sample size calculation is missing. How did the authors calculate the sample size? Inclusion and exclusion criteria is unclear. Why patients with mental disorders or other diseases associated with eating disorders did not exclude from the study?
Results
Table 3 is confusing. Odds ratio in logistic regression is estimated via Exp (B); it seems that there were some mistakes regarding the calculation of OR e.g. (Exp (-0.45) = 0.64. Table 3: 95% CI is related to odds ratio or β? It is unclear. Statistical analyses needs to check by a statistician. Authors should focus on results of multiple regression.
Discussion
Discussion section is very long. It needs to summarize. The authors should add probable mechanisms in the association between age and BMI with eating disorders.
Author Response
Dear Reviewer,
Please see the attachment.
Regards
Stephanie Pirotta
